# Assessment of Systemic and Periodontal Conditions in Pregnant Women and Their Impact on Neonatal Birth Weight: A Prospective Cohort Study

**DOI:** 10.3390/ijerph22030355

**Published:** 2025-02-27

**Authors:** Bruno Gualtieri Jesuino, Gerson Aparecido Foratori-Junior, Ana Virgínia Santana Sampaio Castilho, Ana Carolina da Silva Pinto, Gabriela de Figueiredo Meira, Sílvia Helena de Carvalho Sales-Peres

**Affiliations:** 1Department of Pediatric Dentistry, Orthodontics and Collective Health, Bauru School of Dentistry, University of São Paulo, Bauru 05508-220, São Paulo, Brazil; brunogjesuino@gmail.com (B.G.J.); anavcastilho@usp.br (A.V.S.S.C.); ana.pin@usp.br (A.C.d.S.P.); gabrielameira@usp.br (G.d.F.M.); 2Department of Biological Sciences, Bauru School of Dentistry, University of São Paulo, Bauru 05508-220, São Paulo, Brazil

**Keywords:** birth weight, quality of life, mothers, oral hygiene, periodontitis, pregnancy

## Abstract

The aim of this study was to assess some variables of women in the 27th week of pregnancy and after childbirth, in addition to determining which of these variables were associated with low birth weight during the coronavirus disease (COVID-19) pandemic. The patients were divided into two groups: mothers with normal-weight babies (G1 = 60) and mothers with below-normal-weight babies (G2 = 16). The variables assessed were education, monthly family income, anthropometric parameters, systemic health, periodontal condition, oral hygiene habits, Oral Health Impact Profile–14 results, data from the babies at birth, and a COVID-19 diagnosis during pregnancy. The mothers in G1 showed greater weight gain during pregnancy. There was an increase in tooth plaque percentage, probing pocket depth, and clinical attachment level during the study period for both groups and an increase in periodontitis cases in the patients from G1. The mothers from G1 had longer pregnancy periods and delivered taller babies with a higher body mass index. A one-unit increase in weight during pregnancy decreases the likelihood of having a below-normal-weight baby by 11.3% [confidence interval = 2.4–20.4%]. Weight gain during pregnancy is a protective factor that decreases the likelihood of babies being born with below-normal weights.

## 1. Introduction

The association between pregnancy and periodontitis has been documented in the literature [1,2]. This association is attributed to increased levels of estrogen and progesterone, resulting in a reduced immune response and, consequently, heightened inflammation of oral tissues [1].

Some studies have indicated a relationship between periodontitis and low birth weight [3,4,5]. Furthermore, the relationship between periodontitis and prematurity has been widely discussed, with studies reporting that pregnant women with periodontitis have an increased risk of premature delivery [6,7]. A recent study found that mothers who gained excessive weight during pregnancy had babies with above-normal weight [8].

Periodontal health and adverse birth outcomes, such as low birth weight or prematurity, are explained by two proposed mechanisms: the direct pathway involving anaerobic and gram-negative bacteria present in the gingival biofilm, which can lead to bacteremia, and the indirect pathway involving pro-inflammatory markers that enter the patient’s bloodstream from the gingival submucosa [9].

Recent studies have assessed the systemic, periodontal, and quality-of-life conditions of obese women during and after pregnancy. It turned out that this group had a higher prevalence of hypertension during pregnancy, periodontitis during and after pregnancy, and a worsening quality of life during pregnancy [10,11]. Notably, some systematic reviews and meta-analyses have shown possible associations between coronavirus disease (COVID-19) during pregnancy and premature birth [12,13]. Therefore, this study aimed to assess variables such as education, monthly family income, anthropometric parameters, systemic health, periodontal condition, oral hygiene habits, and quality of life of women in the 27th week of pregnancy and after childbirth, in addition to determining which of these variables were associated with low birth weight during the COVID-19 pandemic, as there has been a shortage of studies of this nature.

## 2. Materials and Methods

This study followed the STROBE guidelines [14] and was a prospective cohort study. The study was approved by the Ethics and Research Committee of Bauru School of Dentistry, University of São Paulo (FOB-USP) (CAAE: 54913822.6.0000.5417).

A total of 129 pregnant women were recruited in 2020 from the Basic Health Units of Bauru City, São Paulo, Brazil. In the first semester of 2022, treatments after childbirth were initiated to facilitate the categorization of groups based on the weight of the infants. Mothers with normal-weight babies (G1 = 60) and mothers with below-normal-weight babies (G2 = 16) were selected based on medical records, resulting in a total of 76 pregnant women. Since the initiation of post-childbirth appointments in 2022, only 29 patients from G1 and seven patients from G2 returned for post-childbirth services. Therefore, 31 patients from G1 and nine from G2 were interviewed via telephone. It is important to highlight that a minimum waiting period of 3 months was required before post-childbirth services were provided, as the hormonal imbalance caused by pregnancy could still be present in their bodies. The patients in this study were met approximately 1–2 years after childbirth.

The patients were classified according to the data obtained from their babies’ medical monitoring cards. The number of samples and their ages were determined by the availability of patients who were recruited at the Basic Health Units and returned for postpartum consultation. The inclusion criteria were as follows: mothers with assessable data from the third trimester of pregnancy (from the 27th week of pregnancy), aged between 18 and 50 years, regularly monitored by an obstetrician during pregnancy and by a pediatrician after childbirth, and who did not necessitate absolute rest. The exclusion criteria included the following: mothers whose infants had a birth weight of ≥4000 g, those with neuromotor alteration, those taking antibiotics or any medication that may interfere with the periodontal condition, and those using alcohol, tobacco, or illegal drugs.

Education level was categorized on a numerical scale ranging from illiteracy to doctoral-level education. Similarly, monthly family income was assessed on a scale from a maximum of R$937.00 to amounts exceeding R$4685.01.

Mothers were considered hypertensive if their blood pressure readings were ≥140/90 mmHg [15]. To confirm the diagnosis of arterial hypertension, the data were checked during pregnancy (T1) on a medical monitoring card and measured during the appointment post-childbirth (T2). Regarding diabetes mellitus, whether or not the diagnoses existed on the medical monitoring card was checked, and anthropometric assessments were made to obtain the body mass index (BMI) before pregnancy and at T2. This process involved measuring the weight and height to calculate the BMI following the standard procedure recommended by the World Health Organization [16]. The parameters of normality for weight gain during pregnancy were based on a protocol proposed by the Institute of Medicine (2009) [17].

The assessments of the periodontal condition of the pregnant women during T1 and T2 were done with the help of a manual North Caroline–type periodontal probe (QD.320.05, Quinelato, Schobell Ind. Ltd., Rio Claro, São Paulo, Brazil) and a number 5 plain oral mirror (Cod. 7503, Duflex/SS White, Juiz de Fora, Minas Gerais, Brazil). All existing teeth were assessed [18], excluding the third molars. The probing pocket depth was assessed based on the distance from the free gingival margin to the most apical bottom point of the periodontal pocket. A periodontal probe was introduced parallel to the tooth axis under light pressure. All teeth were evaluated at six sites: mesial, central, and distal of the buccal and mesial, central, and distal of the lingual [18]. Recession and gingival hyperplasia were considered in the calculation of clinical attachment levels. Gingival bleeding after probing was evaluated using the index proposed by Ainamo and Bay in 1975 [19]. Bleeding was considered positive when bleeding occurred 10 s after probing. A patient with an intact periodontium (without pockets associated with clinical attachment loss) was diagnosed with a “case of gingivitis”, according to the percentage of sites with bleeding on probing (BOP) of ≥10%. This condition was further categorized as local (percentage of sites with BOP ≥10% and ≤30%) or general (percentage of sites with BOP > 30%) [20].

The patients were assessed in T1 and T2 in relation to the presence and absence of periodontitis following the criteria of Tonetti et al. They were diagnosed with periodontitis if (i) interdental clinical attachment loss (CAL) was observed at two or more non-adjacent teeth, or (ii) buccal or oral CAL of ≥3 mm with probing pocket depth of >3 mm was observed at two or more teeth. It is important to note that this CAL is not related to non-periodontal causes, as described by Tonetti et al. [21]. It was not possible to evaluate the postpartum periodontal parameters in the patients who were interviewed via telephone.

Quality of life was measured at T1 and T2 using the adapted and validated Oral Health Impact Profile-14 (OHIP-14) questionnaire, which was applied by means of standardized interviews so that different interpretations could be avoided among patients, minimizing the subjectivity of the questionnaire. The OHIP-14 is an adapted version of the OHIP-49, where the assessed dimensions of impact are functional limitation, physical pain, psychological discomfort, physical disability, psychological disability, social disability, and handicap; two questions were applied for each dimension [22]. The answer codes were as follows: 0 = never, 1 = rarely, 2 = sometimes, 3 = frequently, and 4 = always. The average score was taken from two questionnaires for each dimension and added together to the value of the seven dimensions. The total value varied from 0 to 28, where the zero result (0) was classified as “no impact over the oral health impact profile”; 0 < OHIP-14 ≤ 9 was classified as “low impact”; 9 > OHIP-14 ≤ 18 was classified as “moderate impact”; and 18 < OHIP-14 ≤ 28 was classified as “high impact” [23].

The baby’s birth weight was obtained from the medical monitoring card. The weight during birth was standardized according to the World Health Organization [24]: low weight = <2500 g; insufficient weight = 2500 to 2999 g; suitable weight = 3000 to 3999 g [25]; and overweight at birth was considered when the weight at birth was ≥4000 g [26]. In other words, values less than or equal to 2.999 kg were considered below-normal weight at birth and those between 3.000 and 3.999 kg were considered normal weight. The height of each baby was measured in centimeters. The infants’ BMI was calculated by dividing the weight (in kilograms) by the square of the height (in meters). A normal birth period was considered to be the one that commenced since the 37th week of pregnancy [24,27,28].

All analyses were performed using the R statistical software (version 4.4.2) [29]. The significance level was set to 5%. Fisher’s exact test was used to study the relationship between two categorical variables, and the remaining tests are described in the table captions.

## 3. Results

### 3.1. Age, Education Level, Family Monthly Income and Gestational Weight Gain

No difference was found in the age of the groups (*p* = 0.956); the average age in G1 and G2 was 30.1 (standard deviation [SD] = 6.20) and 30.2 (SD = 5.3), respectively. There were also no differences between the groups in terms of education (*p* = 0.55) or monthly family income (*p* = 0.909). Statistically significant differences were observed between the groups in weight gain during pregnancy (*p* = 0.006), and the averages for G1 and G2 were 12.9 (SD = 7) and 8.4 (SD = 5), respectively. According to the recommended weight gain as per the Institute of Medicine (2009) [17], with pre-pregnancy maternal BMI, 26 (43.3%) patients in G1 presented with excessive weight gain during pregnancy, and 34 (56.7%) presented with normal weight gain. In G2, three (18.8%) presented with excessive weight gain and the other 13 (81.2%) presented with normal weight gain.

### 3.2. Arterial Hypertension and Diabetes Mellitus

During pregnancy, 53 (88.3%) patients in G1 did not have arterial hypertension and 7 (11.7%) did. However, in G2, 13 (81.2%) patients did not have hypertension and three (18.8%) had hypertension. During post pregnancy, 58 patients (96.7%) in G1 did not have hypertension, and 2 (3.3%) had hypertension. In contrast, among the patients in G2, 14 (87.5%) did not have hypertension and two (12.5%) did. When the evolution of this variable was assessed over time, that is, from pregnancy to post childbirth, a statistically significant difference was observed for G1 (*p* = 0.012) and G2 (*p* = 0.025). When diabetes mellitus was evaluated, no statistically significant differences were observed between the groups during pregnancy (*p* = 0.431) or after birth (*p* = 0.513). During pregnancy, among the patients from G1, 53 (88.3%) did not have this condition, while seven (11.7%) did; nonetheless, among the patients from G2, 13 (81.2%) had the condition, while three (18.8%) did not. However, during the post-childbirth period, in G1, 58 (96.7%) did not have diabetes mellitus and two (3.3%) did, while in G2, 15 (93.8%) did not have the condition and one (6.2%) did. When the evolution of this variable was assessed from pregnancy to post-birth, a statistically significant difference was observed for G1 (*p* = 0.012) but not for G2 (*p* = 0.187).

### 3.3. Oral Hygiene Habits and Periodontal Condition

When the variables related to oral hygiene habits and periodontal condition were evaluated inside the same period, no statistically significant difference could be noticed in any comparisons of these variables.

When the evolution of oral habits and periodontal conditions was studied over time in each group, an increase in tooth plaque percentage was observed in G1 (*p* = 0.005) and G2 (*p* = 0.029). A statistically significant difference was observed in the average probing pocket depth (average) for G1 (*p* = 0.004) and G2 (*p* = 0.027), as well as in the clinical attachment level (average) in G1 (*p* < 0.001) and G2 (*p* = 0.036) and the periodontitis variable (yes or no) in G1 (*p* = 0.009). However, there were no differences related to periodontitis (yes or no) in G2 (*p* = 1.0). It is noticeable that, in G1, all patients with periodontitis during pregnancy maintained the condition after childbirth, and nine who did not present with this condition during pregnancy started to present with it after childbirth. Table 1 and Table 2 summarize these events.

### 3.4. OHIP-14

When the variables related to quality of life were evaluated during the same period, statistically significant differences were noted only when comparing functional limitations during pregnancy, showing a greater impact of G2 on this dimension. Table 3 compares these dimensions.

When the evaluation of these dimensions and the overall OHIP-14 score were studied over time within each group, a statistically significant difference was observed in G1 for the psychological discomfort dimension (*p* = 0.01) but not in G2 (*p* = 0.395). In contrast, statistically significant differences were observed in the physical disability dimension for G2 (*p* = 0.025) but not for G1 (*p* = 0.05).

### 3.5. Babies’ Data

When the babies’ data were evaluated, statistically significant differences were observed between the groups in terms of height at birth, week of birth, and BMI at birth; the mothers in G1 had taller babies with greater BMI and a greater number of weeks of pregnancy, as seen in Table 4. Notably, one patient from G1 (28 weeks) and two from G2 (34 and 36 weeks) presented with premature birth.

Logistic regression models were adjusted to study the relationship between a set of variables and the weight of the baby (normal × below normal) [30]. These models aim to show the chances of a patient with certain characteristics (variable values) having a normal-weight baby. Only the patients who presented with values for all variables considered in the model were included in the adjustment of the model. Thus, as there were a lot of missing data in the variable “periodontitis during pregnancy (yes/no)”, this variable was removed from the model, and the same model was adjusted using data from 76 patients. In this new model, weight gain during pregnancy was statistically significant (*p* = 0.02), and when weight during pregnancy increased by one unit, the chance of having a baby with a weight below normal was reduced by 11.3% (confidence interval [CI] = 2.4–20.4%, Table 5).

### 3.6. COVID-19 Diagnosis During Pregnancy

Regarding COVID-19 diagnosis during pregnancy, six patients from G1 and one from G2 presented with the disease during pregnancy.

## 4. Discussion

A study in which the independent variable was weight gain during pregnancy showed that mothers presenting excessive weight gain during pregnancy had babies with a BMI above normal at birth, even when the mothers in this group presented with a greater prevalence of hypertension and periodontitis during pregnancy [8]. It is interesting to compare the results of the study by Jesuino et al. (2020) [8] with those of this study, since, in the final logistic regression model, weight gain during pregnancy proved to be statistically significant (*p* = 0.02); if weight increases by one unit during pregnancy, the chance of having a baby with a below-normal weight decreases by 11.3%, with a CI of 2.4–20.4%. Therefore, this result completes, in a way, the results found by Jesuino et al., [8] since the mothers who gained a lot of weight during pregnancy had babies with BMI considered above normal at birth and not below it. Therefore, it is important to highlight that pregnant women must have proper medical monitoring and health orientation by a multidisciplinary team to facilitate adequate weight gain.

No significant difference in age was observed between the groups (*p* = 0.956). Moreover, no significant differences were observed between the groups in terms of education (*p* = 0.55) or monthly family income (*p* = 0.909). Although no difference was observed in this study, it is interesting to highlight that previous literature shows that women with lower monthly family income tend to consume cheaper foods, which are consequently more caloric and less nutritious; this tends to make these women gain a lot of weight during pregnancy, which impacts the baby’s weight at birth [8,31].

In a study where the independent variable was gestational hypertension, it was associated with obesity, and the combination of these variables seems to be related to the worsening of periodontal parameters. Therefore, there is a worsening of women’s quality of life [32]. In a cross-sectional study of 80 patients, 40 of whom had diabetes mellitus during pregnancy and 40 did not, showed that women with this pathology presented with a higher severity of periodontitis, lower socioeconomic status, higher degree of overweight/obesity, and greater risk of prematurity. Socioeconomic and cultural status and BMI are significant predictors of periodontitis, and diabetes mellitus during pregnancy is a predictor of prematurity [33].

A systematic review and meta-analysis showed that in studies not using insulin, women with gestational diabetes mellitus had a higher chance of some conditions, such as cesarean section, premature birth, a low 1 min Apgar score (used to assess the health condition of the newborn), macrosomia, and a newborn larger than ideal for gestational age. In studies using insulin, the chances of having a baby large for gestational age or with respiratory distress syndrome or neonatal jaundice or requiring admission to the neonatal intensive care unit were higher in women with gestational diabetes mellitus than in those without diabetes [34].

When oral hygiene habits and periodontal parameters were assessed, an increase in the percentage of tooth plaque was observed in G1 (*p* = 0.005) and G2 (*p* = 0.029). It is interesting to note that, although there were no differences between the two groups during the periods, with respect to daily tooth brushing and use of dental floss, both groups showed an increase in the percentage of tooth plaque, which can be explained by the new routine of the woman who has to look after her baby and ends up not spending the necessary quality time on her own oral hygiene [8,32].

A recent study demonstrated that periodontitis during pregnancy is associated with high BMI, excessive weight gain, lower socioeconomic status, worse oral hygiene, and a higher impact on quality of life [31]. Other studies have suggested an association between periodontitis and low birth weight [3,4,5] and between periodontitis and prematurity [6,7].

A study showed that overweight and obese pregnant women in the Brazilian public health system presented with a higher prevalence of arterial hypertension, a worse periodontal condition, and worse quality of life [10]. In obese women, periodontitis and hypertension are highly prevalent during pregnancy and can result in worse quality of life [11]. In women with gestational hypertension, worsening of some parameters, such as physical pain, physical disability, psychological disability, and psychological discomfort during pregnancy and post childbirth, was observed [32]. In pregnant women, periodontitis had a strong impact on quality of life in all dimensions assessed [31].

When the babies’ birth data were evaluated, a significant difference was observed between the groups in the following variables: height at birth (*p* < 0.001), week of birth (*p* < 0.001), and BMI at birth (*p* < 0.001); that is, the mothers in G1 had taller babies, with higher BMI and a greater number of gestational weeks, which was somewhat expected because the babies’ birth weight was the independent variable in this study. Two mechanisms have been proposed to explain the birth of babies with low birth weight and premature births. The first is the direct pathway, which is related to the presence of anaerobic Gram-negative bacteria present in the gingival biofilm, while the second is the indirect pathway, in which pro-inflammatory markers are produced, which enter the bloodstream from the gingival submucosa. Both mechanisms result in inflammation, which is mediated by the inflammatory immune response and suppression of growth factors in the placenta [9,35].

Regarding the independent variable of COVID-19 diagnosis during pregnancy, six patients in G1 and one patient in G2 presented with the disease during the gestational period. All patients who presented with this condition had babies without a premature birth, and only one had a birth weight below normal. Despite the limitations of the small sample size of patients with a positive diagnosis of the disease during pregnancy in this study, the patients differed from those in a systematic review and meta-analysis that concluded that gestational COVID-19 was associated with a higher rate of premature births [12]. Another systematic review and meta-analysis found that severe acute respiratory syndrome coronavirus 2 infection during pregnancy was associated with the risk of preeclampsia, stillbirth, premature birth, and admission to the neonatal intensive care unit [13]. It has also been reported that viral infection during pregnancy is dangerous, especially in the third trimester, which increases the risk of placental damage that can lead to spontaneous abortion, premature birth, and stillbirth [36]. It is important to highlight that some studies have shown that COVID-19 is asymptomatic in many pregnant women [37,38], which may have caused a bias in the present study because some patients may have had the disease during pregnancy and not even been aware of it.

This study has some limitations due to the difficulty encountered in data collection. The first limitation is related to the small sample size of G2, since most of the patients who agreed to participate in the study had babies with a weight considered normal at birth. Some women refused to participate in the study due to the deaths of their babies, and their decision to not participate was respected. Furthermore, regarding the collection of periodontal data, mothers experienced great difficulty in returning for postpartum consultations since many reported difficulties in the new routine of caring for the baby, in addition to financial and work difficulties. Despite these limitations, this study contributes to the literature by evaluating patients during pregnancy and after childbirth, showing how the new maternal care routine affects the oral health and hygiene habits of these mothers after childbirth, thereby showing the impact of their oral health on their quality of life. In addition, this study shows the importance of holistic patient care, as understanding the possible worsening of behavioral habits due to the new routine in relation to oral health and its harms, such as increased plaque, probing pocket depth, clinical attachment level, and periodontitis, makes it possible to guide and treat these patients. In addition, it demonstrates the importance of systemic healthcare, especially in relation to adequate gestational weight gain.

## 5. Conclusions

It was concluded that G1 presented with excessive weight gain during pregnancy, and an increase in systemic health and worsening of oral health was observed in both groups. There were differences between the groups in terms of quality of life with respect to the psychological discomfort dimension in G1 as well as functional limitations during pregnancy and physical disability in G2. The G1 mothers had taller babies with higher BMI and had more weeks of pregnancy; however, gestational weight gain within the recommended parameters was a protective factor to reduce the chance of the baby being born below normal weight.

## Figures and Tables

**Table 1 ijerph-22-00355-t001:** Frequency of the periodontitis variables during pregnancy (yes-no) and post-pregnancy (yes-no) in the sample for G1.

Total (Post-Birth)	Yes (Post-Birth)	No (Post-Birth)	
17 (58.6%)	9 (52.9%)	8 (47.1%)	No (Pregnancy)
12 (41.4%)	12 (100%)	0 (0%)	Yes (Pregnancy)
29 (100%)	21 (72.4%)	8 (27.6%)	Total (Pregnancy)

Source: Author.

**Table 2 ijerph-22-00355-t002:** Descriptive statistics of periodontal variables by group variable in the sample.

*p*	G2 (n = 7)	G1 (n = 29)		
	Median	Median		
	[1st–3rd Quartiles]	[1st–3rd Quartiles]		
0.453 *	58.3	41.6		Gingival bleeding post probing during pregnancy (%)
	[48.8–60.1]	[28.0–57.1]		
0.88 *	54.3	53.2		Gingival bleeding post probing post birth (%)
	[46.1–67.1]	[30.8–72.6]		
0.507 *	58.3	61.5		Tooth plaque during pregnancy (%)
	[57.1–72.3]	[50.0–78.6]		
0.311 ^¢^	78.8	80.3		Tooth plaque post birth (%)
	[70.5–100.0]	[57.1–100.0]		
0.368 ^¢^	2.2	2.1		Probing pocket depth during pregnancy (mm)
	[2.0–2.5]	[2.0–2.2]		
0.549 ^¢^	2.2	2.2		Probing pocket depth post birth (mm)
	[2.1–2.4]	[2.1–2.3]		
0.46 ^¢^	2.2	2.2		Clinical attachment level during pregnancy (mm)
	[2.1–2.5]	[2.0–2.3]		
0.436 ^¢^	2.4	2.2		Clinical attachment level post birth (mm)
	[2.2–2.4]	[2.1–2.4]		
0.219	2 (28.6%)	17 (58.6%)	No	Periodontitis during pregnancy
	5 (71.4%)	12 (41.4%)	Yes	
1.0	2 (28.6%)	8 (27.6%)	No	Periodontitis post birth
	5 (71.4%)	21 (72.4%)	Yes	

Source: Author; ^¢^ Wilcoxon. * T test.

**Table 3 ijerph-22-00355-t003:** Descriptive statistics about the OHIP-14 dimensions by the group variable in the sample.

*p*	G2 (n = 16)	G1 (n = 60)	
	Median	Median	
	[1st–3rd Quartiles]	[1st–3rd Quartiles]	
0.004 ^¢^	0.0	0.0	Functional limitation during pregnancy
	[0.0–1.0]	[0.0–0.0]	
0.084 ^¢^	0.0	0.0	Functional limitation post birth
	[0.0–1.1]	[0.0–0.0]	
0.392 ^¢^	1.5	1.0	Physical pain during pregnancy
	(0.9–2.6)	(0.0–2.0)	
0.232 ^¢^	2.0	1.0	Physical pain post birth
	(0.4–3.6)	(0.0–3.0)	
0.126 ^¢^	1.5	0.5	Psychological discomfort during pregnancy
	(0.4–4.0)	(0.0–2.5)	
0.18 ^¢^	3.2	1.2	Psychological discomfort post birth
	(0.8–4.0)	(0.0–3.5)	
0.378 ^¢^	0.0	0.0	Physical disability during pregnancy
	(0.0–0.6)	(0.0–0.0)	
0.105 ^¢^	0.5	0	Physical disability post birth
	(0.0–3.2)	(0.0–1.0)	
0.156 ^¢^	0.5	0.0	Psychological disability during pregnancy
	(0.0–2.1)	(0.0–1.5)	
0.172 ^¢^	1.2	0.0	Psychological disability post birth
	(0.0–2.2)	(0.0–2.0)	
0.341 ^¢^	0.0	0.0	Social disability during pregnancy
	(0.0–0.8)	(0.0–1.5)	
0.087 ^¢^	0.2	0.0	Social disability post birth
	(0.0–2.0)	(0.0–0.6)	
0.262 ^¢^	0.0	0.0	Handicap during pregnancy
	(0.0–1.2)	(0.0–0.6)	
0.219 ^¢^	0.0	0.0	Handicap post birth
	(0.0–2.0)	(0.0–0.6)	
0.275 ^¢^	4.0	4.0	Overall score during pregnancy
	(1.4–12.4)	(0.0–7.6)	
0.078 ^¢^	8.2	3.0	Overall score post birth
	(3.5–15.9)	(0.0–11.1)	
0.244			OHIP-14 categorization during pregnancy
	1 (6.2%)	3 (5.0%)	High impact
	6 (37.5%)	10 (16.7%)	Moderate impact
	7 (43.8%)	30 (50.0%)	Low impact
	2 (12.5%)	17 (28.3%)	No Impact
	16 (21.1%)	60 (78.9%)	Overall
0.312			OHIP-14 categorization post birth
	3 (18.8%)	5 (8.3%)	High impact
	5 (31.2%)	14 (23.3%)	Moderate impact
	5 (31.2%)	17 (28.3%)	Low impact
	3 (18.8%)	24 (40.0%)	No Impact
	16 (21.1%)	60 (78.9%)	Overall

Source: Author; Caption: ^¢^ Wilcoxon. OHIP, Oral Health Impact Profile–14.

**Table 4 ijerph-22-00355-t004:** Descriptive statistics of the babies’ variables at birth by the group variable in the sample.

*p*	G2 (n = 16)	G1 (n = 60)	
	Median	Median	
	[1st–3rd Quartiles]	[1st–3rd Quartiles]	
<0.001 ^¢^	0.47	0.49	Height at birth
	[0.45–0.47]	[0.48–0.5]	
<0.001 ^¢^	38	40	Week of birth
	[37–39]	[39–40]	
<0.001 ^¢^	12.6	14.1	BMI at birth
	(12.2–13.0)	(13.4–14.9)	

Source: author; Caption: ^¢^ Wilcoxon. BMI, body mass index.

**Table 5 ijerph-22-00355-t005:** Logistic regression models for the chance of a baby to be born with weight below normal without the periodontitis during pregnancy variable.

UL	LL	Odds Ratio	*p*	T Value	Standard Error	Estimation	Category	Variable
			0.278	1.085	2.573	2.790	Intercept	
1.03	0.812	0.92	0.167	−1.383	0.060	−0.083		BMI during pregnancy
0.976	0.796	0.887	0.020	−2.331	0.051	−0.119	Weight gain during pregnancy	
23.101	0.533	3.585	0.171	1.368	0.933	1.277	Yes	Arterial hypertension during pregnancy
9.223	0.263	1.675	0.560	0.583	0.884	0.516	Yes	Diabetes mellitus during pregnancy
1.086	0.877	0.981	0.717	−0.363	0.053	−0.019		Age

Source: Author. BMI, body mass index; LL, lower limit; UL, upper limit.

## Data Availability

The data from this study are in the possession of the research team.

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
