# Peer review of "Assessment of Systemic and Periodontal Conditions in Pregnant Women and Their Impact on Neonatal Birth Weight: A Prospective Cohort Study"

_ijerph, 2025, doi:10.3390/ijerph22030355_

Round 1

Reviewer 1 Report

Comments and Suggestions for Authors

This study reports a level of  oral status in view of periodontal condition, oral hygiene assessment and quality of life of women from the 27th pregnancy and after the birth of the babies, during COVID-19 20 pandemic.

 It is a well controlled prospective study using  pregnant women. The study is well written and the methods used for the collection and analysis of the data are standard and reliable.  The statistical analysis is correctly applied. The authors may find the following comments

a/Keywords: please add weight at birth, pregnancy, oral hygiene as a part of measured biological factors

b/Patients and Methods:

1/ could you indicate inclusion and exclusion criteria in table?

2/ in both groups patients were in age between 18 and 50 years old, I would suggest to add statistics according to age of women, how the age  may influence the results

3/ in tables results are recorded in  numbers with commas, please replace with dots

3a/ Periodontal indexes which were measured I would suggest to show results and statistical differences

4/ In discussion, please add a strong and weak points of your clinical trial.

 5/ conclusions are well presented

Author Response

This study reports a level of  oral status in view of periodontal condition, oral hygiene assessment and quality of life of women from the 27th pregnancy and after the birth of the babies, during COVID-19 20 pandemic.

 It is a well controlled prospective study using  pregnant women. The study is well written and the methods used for the collection and analysis of the data are standard and reliable.  The statistical analysis is correctly applied. The authors may find the following comments

a/Keywords: please add weight at birth, pregnancy, oral hygiene as a part of measured biological factors

Added: birth weight, pregnancy, oral hygiene.

b/Patients and Methods:

1/ could you indicate inclusion and exclusion criteria in table?

We appreciate the suggestion, but after talking to the authors we thought it would be better to keep the inclusion and exclusion criteria within the text.

2/ in both groups patients were in age between 18 and 50 years old, I would suggest to add statistics according to age of women, how the age  may influence the results

The number of samples and their ages were determined by the availability of patients who were recruited at the Basic Health Units and returned for postpartum consultation. (information included in the article).

3/ in tables results are recorded in  numbers with commas, please replace with dots

The suggestion was accepted.

3a/ Periodontal indexes which were measured I would suggest to show results and statistical differences

New table was added to resolve this problem.

4/ In discussion, please add a strong and weak points of your clinical trial.

The suggestion was accepted.

 5/ conclusions are well presented

Thank you very much.

Reviewer 2 Report

Comments and Suggestions for Authors

Dear authors

i have some important comments reagrding your manuscript :

- the quality of english could be improuved and shorten sentences should be used to provide an easier lecture

- the introduction is too short and globally the number of references (26) is too short for this type of paper

-the main remark concerns the number of patients in the 2 groups  : you have only 16 patients in G2 which is quite limited especially if you consider that you lost 9 patients for the second examination post birth.

With only 7 patients in group2 post birth, any stastical analysis could not appear as significant

Comments on the Quality of English Language

All the sentences are too long

the global quality of the english should be improved

Author Response

Dear authors

i have some important comments reagrding your manuscript :

- the quality of english could be improuved and shorten sentences should be used to provide an easier lecture

The article was reviewed by a company specializing in English language reviews.

- the introduction is too short and globally the number of references (26) is too short for this type of paper

The introduction has been extended and the total number of references has increased to 38.

-the main remark concerns the number of patients in the 2 groups  : you have only 16 patients in G2 which is quite limited especially if you consider that you lost 9 patients for the second examination post birth.

With only 7 patients in group2 post birth, any stastical analysis could not appear as significant.

This comment really makes a lot of sense, but it was very complicated to collect these 16 patients during these periods (during pregnancy and post-birth), given their specificity (being treated during pregnancy, in addition to postpartum and also having a baby with low birth weight). However, these observations of the low sample in G2 are described in the article as a limitation of the study.

Reviewer 3 Report

Comments and Suggestions for Authors

I would like to congratulate the authors for their effort in conducting a prospective cohort study, I think the manuscript is well written, however after full reading, I think the title is inappropriate in relation to all the variables evaluated throughout the study.

 The following comments are addressed and require some modifications to enhance the quality of the manuscript:

 In abstract authors start with the objective of the study:

 “The aim of this study was to assess the systemic and periodontal conditions and quality of life of women from the 27th pregnancy and after the birth of the babies, during COVID-19 pandemic.”

However, throughout the manuscript authors do not relate or explore the relationship between systemic and periodontal conditions in pregnant women/covid 19 positive, so I suggest to remove this variable from the title, as it does not add information and may confuse readers. At the end of introduction, the authors recall the aim of the study and  there is also no reference to covid 19.

Introduction, in my opinion, is short and does not focus the main subjects related to the manuscript (systemic and periodontal conditions/pregnant women/prematurity and low birth weight/covid 19).  I think essential you mention the innovation and justification of this study in relation to previous studies.

Once again, I ask the authors to clarify the objectives of the study and to adequate the manuscript title to this.

“ The aim of this study was to assess the systemic and periodontal conditions and quality 19 of life of women from the 27th pregnancy and after the birth of the babies, during COVID-19  pandemic”.-abstract

the proposition of this study is to assess the relationship between periodontitis, below normal weight at birth and prematurity.” -introduction

In Materials and Methods section, the methodology is not clear, and I have some concerns in some aspects:

“Therefore, 31 patients from G1 and 9 from G2 were interviewed by telephone.”- How did you evaluate the postpartum periodontal parameters in these patients?

Authors must mention the existence or not of drop outs; please explain T1 and T2, the first time it appears in text.

How did you calculate the sample size?

Results are not quite clearly presented, to me, especially in periodontal parameters, it should be more explicit in text e.g. which group has worse pocket depth or higher CAL. Table 4 also needs a brief explanation through text.

Discussion is brief and made in relation to 4 studies after 2020 and 2 systematic reviews, one from 2013 and one from 2019. I believe that there is more up-to-date literature, that could had been included, to make discussion more enriching, with more current data, regarding the relationship between systemic (diabetes and hypertension) / periodontal conditions and the weight of baby at birth, which corresponds to the title of the manuscript.

 Conclusions should be short and respond to the objectives proposed.

Author Response

I would like to congratulate the authors for their effort in conducting a prospective cohort study, I think the manuscript is well written, however after full reading, I think the title is inappropriate in relation to all the variables evaluated throughout the study.

 The following comments are addressed and require some modifications to enhance the quality of the manuscript:

 In abstract authors start with the objective of the study:

 “The aim of this study was to assess the systemic and periodontal conditions and quality of life of women from the 27th pregnancy and after the birth of the babies, during COVID-19 pandemic.”

However, throughout the manuscript authors do not relate or explore the relationship between systemic and periodontal conditions in pregnant women/covid 19 positive, so I suggest to remove this variable from the title, as it does not add information and may confuse readers. At the end of introduction, the authors recall the aim of the study and  there is also no reference to covid 19.

It was changed by: “The aim of this study was to assess the variables of women in the 27th week of pregnancy and after childbirth, in addition to determining which of these variables are associated with low birth weight during the coronavirus disease (COVID-19) pandemic. The patients were divided into two groups: mothers with normal-weight babies (G1 = 60) and mothers with below-normal-weight babies (G2 = 16). The variables assessed were education...”

The new title is: Assessment of Systemic and Periodontal Conditions in Pregnant Women and Their Impact on Neonatal Birth Weight: A Prospective Cohort Study.

Now COVID-19 is discussed in the article.

Introduction, in my opinion, is short and does not focus the main subjects related to the manuscript (systemic and periodontal conditions/pregnant women/prematurity and low birth weight/covid 19).  I think essential you mention the innovation and justification of this study in relation to previous studies.

Indroduction was reformulated.

Once again, I ask the authors to clarify the objectives of the study and to adequate the manuscript title to this.

“ The aim of this study was to assess the systemic and periodontal conditions and quality 19 of life of women from the 27th pregnancy and after the birth of the babies, during COVID-19  pandemic”.-abstract

“the proposition of this study is to assess the relationship between periodontitis, below normal weight at birth and prematurity.” -introduction

It was changed by: “Therefore, this study aimed to assess variables such as education, monthly family income, anthropometric parameters, systemic health, periodontal condition, oral hygiene habits, and quality of life of women in the 27th week of pregnancy and after childbirth, in addition to determining which of these variables are associated with low birth weight during the COVID-19 pandemic, as there has been a shortage of studies of this nature.”

In Materials and Methods section, the methodology is not clear, and I have some concerns in some aspects:

“Therefore, 31 patients from G1 and 9 from G2 were interviewed by telephone.”- How did you evaluate the postpartum periodontal parameters in these patients?

 It was not possible to evaluate the postpartum periodontal parameters in patients who were interviewed via telephone. (information included in the article).

Authors must mention the existence or not of drop outs; please explain T1 and T2, the first time it appears in text.

The suggestion was accepted.

How did you calculate the sample size? The number of  sample and their ages were in accordance with the availability of patients who were recruited at the Basic Health Units and who returned for consultation after giving birth. (information included in the article).

Results are not quite clearly presented, to me, especially in periodontal parameters, it should be more explicit in text e.g. which group has worse pocket depth or higher CAL.

New table was added to resolve this problem.

Table 4 also needs a brief explanation through text.

A brief explanation was added.

Discussion is brief and made in relation to 4 studies after 2020 and 2 systematic reviews, one from 2013 and one from 2019. I believe that there is more up-to-date literature, that could had been included, to make discussion more enriching, with more current data, regarding the relationship between systemic (diabetes and hypertension) / periodontal conditions and the weight of baby at birth, which corresponds to the title of the manuscript.

12 references were included in the article, especially in the discussion.

 Conclusions should be short and respond to the objectives proposed.

It is not possible to reduce since there would be a lack of information to be concluded, the authors tried, but it was not possible.

Round 2

Reviewer 1 Report

Comments and Suggestions for Authors

The manuscript improved  information needed to follow academic discussion in publication, it can qualified to be accepted

Author Response

The manuscript improved information needed to follow academic discussion in publication, it can qualified to be accepted

The authors are grateful for the suggestions made and would like to thank you for the final recognition.

Reviewer 3 Report

Comments and Suggestions for Authors

I thank the authors for the changes made in manuscript,I think it has improved the quality with more references, new data and a more complete and interesting discussion I would only suggest this small alteration in abstract. After this modification I think the article is suitable for publication.

"The aim of this study was to assess some variables of women in the 27th week of pregnancy and after childbirth, in addition to determining which of these variables are associated with low birth weight during the coronavirus disease (COVID-19) pandemic. The variables assessed were education, monthly family income, anthropometric parameters, systemic health, periodontal condition, oral hygiene habits, Oral Health Impact Profile–14 results, data from babies at birth, and COVID-19 diagnosis during pregnancy. 
The patients were divided into two groups: mothers with normal-weight babies (G1 = 60) and mothers with below-normal-weight babies (G2 = 16)."

Author Response

I thank the authors for the changes made in manuscript,I think it has improved the quality with more references, new data and a more complete and interesting discussion I would only suggest this small alteration in abstract. After this modification I think the article is suitable for publication.

"The aim of this study was to assess some variables of women in the 27th week of pregnancy and after childbirth, in addition to determining which of these variables are associated with low birth weight during the coronavirus disease (COVID-19) pandemic. The variables assessed were education, monthly family income, anthropometric parameters, systemic health, periodontal condition, oral hygiene habits, Oral Health Impact Profile–14 results, data from babies at birth, and COVID-19 diagnosis during pregnancy. 
The patients were divided into two groups: mothers with normal-weight babies (G1 = 60) and mothers with below-normal-weight babies (G2 = 16)."

The suggestion was accepted. The authors would like to thank you for the new suggestion and final recognition.